# Cognitive Function, Healthy Lifestyle, and All-Cause Mortality among Chinese Older Adults: A Longitudinal Prospective Study

**DOI:** 10.3390/nu16091297

**Published:** 2024-04-26

**Authors:** Huiwen Li, Yi Zheng, Qi Li, Mengying Wang

**Affiliations:** 1China Population and Development Research Center, Beijing 100081, China; hli@bjmu.edu.cn; 2Department of Nutrition and Food Hygiene, School of Public Health, Peking University, Beijing 100191, China; 3Institute of Social Development, Chinese Academy of Macroeconomic Research, Beijing 100038, China; q@bjmu.edu.cn; 4Key Laboratory of Epidemiology of Major Diseases, Ministry of Education, Peking University, Beijing 100191, China

**Keywords:** cognitive function, healthy lifestyle, healthy diet, mortality, older adults

## Abstract

Background: Both cognitive decline and unhealthy lifestyles have been linked to an elevated risk of mortality in older people. We aimed to investigate whether a healthy lifestyle might modify the association between cognitive function and all-cause mortality in Chinese older populations. Methods: The final analysis included 5124 individuals free of dementia, selected from the Chinese Longitudinal Healthy Longevity Survey from 2011 to 2018. Cognitive function was assessed in 2011 using the Mini-Mental State Examination (MMSE). A lifestyle score was calculated based on five lifestyle factors, including smoking, alcohol consumption, physical activity, diet, and body mass index. Cox proportional hazards models were performed to evaluate the association between baseline cognitive function and the risk of all-cause mortality, with an interaction term of cognitive function and lifestyle score being added to the models. Results: The average age of participants was 81.87 years old at baseline. During a median follow-up of 6.4 years, 1461 deaths were documented. Both higher cognitive function (HR: 0.96; 95% CI: 0.96–0.97) and a healthier lifestyle (HR: 0.92; 95% CI: 0.87–0.97) were significantly associated with a reduced risk of mortality. We found that lifestyle significantly modified the association of cognitive function with mortality (*p* for interaction = 0.004). The inverse relation between cognitive function and mortality was found to be more pronounced among participants with a healthier lifestyle. Of note, among the lifestyle scores component, diet showed a significant interaction with mortality (*p* for interaction = 0.003), and the protective HR of the all-cause mortality associated with higher MMSE scores was more prominent among participants with healthy diets compared with unhealthy diets. Conclusions: Our study indicates that cognitive decline is associated with a higher risk of mortality, and such associations are attenuated by maintaining a healthy lifestyle, with a particular emphasis on healthy diet.

## 1. Introduction

Dementia is one of the major predictors of mortality in old age [1,2]. It is estimated that by 2050, there will be 131.5 million individuals living with dementia [3], presenting a major challenge for health care and social support systems. Cognitive decline, which worsens progressively with age, has been shown to be the most prevalent cause of dementia [4,5]. Numerous previous prospective studies have demonstrated that lower cognitive function was associated with an increased risk of mortality in middle-aged and older populations [6,7,8,9]. Thus, the prevention and intervention of cognitive decline in older individuals is paramount for longevity and later-life quality. Fortunately, cognitive decline is potentially mutable and preventable through various established contributing factors. 

Notably, emerging evidence has linked various lifestyle factors with cognitive function [10,11,12]. For example, population-based and experimental studies have identified a beneficial effect of physical activity on cognition and brain function [13,14]. In addition to dietary patterns, high nutrient adequacy has been found to be associated with better cognitive performance [15]. Previous studies have indicated that smoking, alcohol consumption, and body mass index (BMI) are also associated with an elevated risk of cognitive decline among old people [16,17]. In addition, it is consistently reported that an unhealthy lifestyle is related to an elevated risk of mortality [18,19,20]. Given their intertwined relationship, it is hypothesized that lifestyle factors may potentially modify the association between cognitive function and all-cause mortality. While several studies have examined relationships of cognitive function and a healthy lifestyle with mortality, the modification effect of a healthy lifestyle on the association between cognition function and the risk of mortality in prospective cohorts is less investigated. 

In the present longitudinal prospective study of the Chinese Longitudinal Healthy Longevity Survey (CLHLS), we sought to verify the association between cognition function and all-cause mortality among Chinese older adults and particularly investigate the modification effects of lifestyle factors, including smoking status, alcohol consumption, physical activity, diet, and BMI, on the association. 

## 2. Materials and Methods

### 2.1. The Study Design and Participants

The CLHLS is an ongoing population-based prospective cohort study, which aims to study determinants of healthy aging among the elderly population in China. The survey applied a multistage, stratified cluster sampling and conducted it in randomly selected counties and cities from 23 of the 31 provinces in Mainland China. Health-related information was collected through structured questionnaires and anthropometric measurements by trained interviewers at each wave of the CLHLS. More detailed information about study design and data collection have been published previously [21,22,23,24].

The present analysis includes data across the three recent waves, each approximately three years apart from the next, from 2011/2012 to 2018. The 2011 survey wave, including 9765 total respondents, was treated as the baseline survey. In the current study, participants who were lost to follow-up or died during 2011–2014 were excluded (*n* = 3699). In accordance with previous studies [25], older adults were defined as those aged 65 years or above. Hence, those younger than 65 years old were excluded (*n* = 57). Moreover, participants with self-reported dementia at baseline (*n* = 67) or those with missing values on cognitive function or lifestyle (*n* = 818) at baseline were excluded, leaving a total of 5124 participants in the final analysis. The detailed study flowchart of participant inclusion and exclusion is provided in Appendix A. Written informed consent was obtained from all participants, and ethical approval was obtained from the Research Ethics Committee of Peking University (IRB00001052–13074).

### 2.2. Measurement of Cognitive Function

The Mini-Mental State Examination (MMSE) continues to be the most widely used instrument in assessing cognitive function [26,27]. Cognitive function was estimated using the validated Chinese version of 24-item MMSE [23,28]. The 24 items cover six dimensions: (1) orientation (5 items), (2) registration (3 items), (3) naming (1 item), (4) attention and calculation (5 items), (5) recall (3 items), and (6) language (7 items), with the total score ranging from 0 to 30. A higher MMSE score indicates better cognitive function. Based on the literature [28], we treated responses of “unable to answer” as “wrong”. Participants were divided into three groups on the basis of the total MMSE score: low ≤ 24; moderate 25–28; high ≥ 29. 

### 2.3. Measurement of Healthy Lifestyle

A lifestyle score was developed based on five factors, which included smoking status, alcohol consumption, physical activity, diet, and BMI, in accordance with previous studies [20,29,30]. Self-reported information on smoking status (never, former, and current), alcohol consumption (never, former, and current), and regular exercise (yes/no) were collected by trained interviewers at baseline. The participants who abstained from smoking and drinking and engaged in regular exercise were defined as healthy. Dietary consumption was assessed using self-reported diversity score according to the World Health Organization recommendations and previous research [31,32]. The respondents were asked to report their current intake frequency of various food groups, including vegetables, fruits, legumes and their products, meat, fish, eggs, milk products, nuts, and tea. In the analysis, participants received a score of 1 point if the response for one food group was ‘almost every day’, ‘once per week at least’, or ‘once per month at least’ and scored 0 point if the response was ‘occasionally’ or ‘rarely or never’. Especially for the frequency of fruit and vegetable intake, participants scored 1 point if the response was ‘almost every day’, ‘almost every day except in winter”, or ‘quite often’; otherwise, they received a score of 0. The dietary diversity score was equal to the sum of the points for all nine food groups mentioned above. The score ranged from 0 to 9, with a higher score indicating better dietary diversity. A healthy diet was defined as the dietary diversity score at or above the mean value in accordance with previous studies [33,34]. Height and weight were measured directly by trained investigators and used to calculate BMI as weight in kilograms divided by height in meters squared. BMI was categorized as underweight (<18.5 kg/m^2^), normal weight (≥18.5 kg/m^2^ and <24.0 kg/m^2^), overweight (≥24.0 kg/m^2^ and <28.0 kg/m^2^), or obese (≥28 kg/m^2^). We defined a healthy body weight as individuals with normal weight. Participants scored 1 point for each favorable behavior (no smoking, no alcohol consumption, regular exercise, healthy diet, and normal weight) and otherwise received a score of 0. The total lifestyle score ranged from 0 to 5, with a higher score indicating a healthier lifestyle. Participants were categorized into three groups according to the total lifestyle score: unhealthy ≤ 2; intermediate = 3; healthy ≥ 4.

### 2.4. Outcome 

The outcome of interest in this analysis was all-cause mortality occurring in the 2014 and 2017/2018 waves. The participants’ vital statistics and date of death were collected from officially issued death certificates whenever available and otherwise through interviews with the next-of-kin. Duration of follow-up was calculated as the time from baseline to death or the censoring time depending on which occurred first.

### 2.5. Covariates

We adjusted socio-demographic characteristics and health status at baseline as potential confounders in the models based on the existing literature [20,30,35,36]. The socio-demographic characteristics were age (65–79/≥80 years), sex (male/female), education (no school/some schooling), residence (urban/rural), marital status (married/other), living arrangement (living with family members/living alone or in an institution), and self-assessment of economic status (rich/so-so/poor). The health statuses were activities of daily living (ADL) and self-reported common chronic diseases diagnosed by a doctor, comprising heart disease, diabetes, cancer, and stroke. ADL was measured using a 0–6 point Katz score scale, including dressing, eating, bathing, continence, toileting and cleaning, and indoor movement [37]. Participants who self-reported experiencing difficulty with any of the ADL tasks were defined as ADL in disability.

### 2.6. Statistical Analyses

Baseline characteristics of the study population were presented as frequency distribution for categorical variables. The characteristics of older adults were compared across different cognitive function groups using the chi-square test. Cox proportional hazards models were constructed to calculate the hazard ratios (HRs) and 95% confidence intervals (CIs) for determining the association between cognitive function and mortality, with low MMSE score group as the reference. Model 1 was adjusted for age and sex. On the basis of Model 1, Model 2 was further adjusted for lifestyle factors. All of the covariates were added in Model 3 based on Model 1, including age, sex, educational levels, residence, marital status, living pattern, self-rated of economic status, ADL in disability, and history of chronic disease (diabetes, heart diseases, cancer, and stroke). Lifestyle and all of the covariates were adjusted in Model 4. The proportional-hazards assumptions for the Cox proportional hazards models were tested using the Schoenfeld residuals method. We conducted the linear trend test by treating MMSE score as a continuous variable. 

In addition, a sensitivity analysis was performed among participants free of heart disease, diabetes, cancer, and stroke at baseline in order to confirm the robustness of our results. 

All analyses were performed using STATA version 14.0 (Stata Corp, College Station, TX, USA). All P values were two-sided with less than 0.05 considered as statistically significant.

## 3. Results

The baseline characteristics of the study participants classified by MMSE scores are shown in Table 1. The average age of participants was 81.87 years old at baseline. Age, gender, education, residence, marital status, self-rated of economic status, ADL disabled, disease history, and lifestyle were significantly different among quintile groups of cognitive function. Participants with lower MMSE scores tended to be older and were more likely to have limited literacy skills. In addition, participants in the low cognitive function group were less likely to be rich, be without ADL disability, and have a healthy lifestyle.

During a median follow-up of 6.4 years, a total of 1461 deaths were observed. We observed a significant association between higher MMSE scores and a reduced risk of mortality, as shown in Table 2. In the model with age and sex being adjusted, a 1-point increase in MMSE scores was associated with a 5% reduction in the risk of mortality (95% CI 4–5%). After further adjustment for age, sex, education, residence, marital status, living pattern, self-rated of economic status, ADL in disability, history of chronic disease (diabetes, heart diseases, cancer, and stroke), and lifestyle, MMSE scores were found to be inversely associated with the risk of mortality. The HR (95% CI) of mortality was 0.96 (0.96–0.97) for a 1-point increase in MMSE scores, and a 39% risk reduction was detected in the highest quintile group compared with the lowest quintile group of MMSE scores (*p* for trend < 0.001). 

In addition, we found that healthier lifestyles were related to a lower risk of mortality (Table 3). In the model with age and sex being adjusted, a 1-point elevation of lifestyle scores was associated with a 13% lower risk of mortality (95% CI 9–17%). With MMSE scores and all covariates being adjusted, healthy lifestyles were associated with a lower risk of death. The HR (95% CI) of mortality was 0.92 (0.87–0.97) for a 1-point increase in lifestyle scores, and a 20% lower risk was detected in the group with healthy lifestyles compared with the group with unhealthy lifestyles (*p* for trend < 0.001).

A stratified association analysis was conducted with the lifestyle scores to examine whether the overall lifestyle modified the association between MMSE scores and all-cause mortality. A significant interaction was found between MMSE scores and lifestyle scores on the risk of mortality (*p* for interaction = 0.004), in which the protective HR of high MMSE scores was stronger among participants who adhered to a healthier lifestyle. The HR (95% CI) of mortality associated with a 1-point increase in MMSE scores was 0.98 (0.96–0.99) among participants with an unhealthy lifestyle, 0.96 (0.95–0.97) among participants with an intermediate lifestyle, and 0.95 (0.93–0.96) among participants with a healthy lifestyle, respectively. When classified into three MMSE score groups, the lower risk of mortality associated with higher MMSE scores was also stronger in healthier lifestyle groups (Figure 1).

A further analysis was conducted to examine the interaction between MMSE scores and each lifestyle behavior separately on the risk of all-cause mortality. Appendix A reported the risks of all-cause mortality per 1-point increase in MMSE scores, stratified by five lifestyle behaviors. Diet showed a significant interaction with MMSE scores for mortality (*p* for interaction = 0.003). A 1-point higher MMSE score was found to have a stronger association with the risk of mortality among participants who maintained a healthy diet (HR 0.956 [95% CI 0.944–0.967]) than those with an unhealthy diet (HR 0.969 [95% CI 0.959–0.979]). In particular, the negative HRs of mortality for the high MMSE score group compared with the low MMSE score group were also diminished among participants who adhered to an unhealthier diet (Figure 2). For other lifestyle behaviors, although a 1-point higher MMSE score also showed stronger associations with the risk of mortality among participants with healthy behavior than those with unhealthy behavior, the interactions did not reach the statistically significant level.

The sensitivity analysis showed that the cognitive–lifestyle interactions remained significant on mortality after excluding participants with diabetes, heart diseases, cancer, or stroke at baseline (Appendix A). 

## 4. Discussion

Maintaining cognitive function and a favorable lifestyle are essential for healthy aging. In this longitudinal prospective study, we found that both cognitive function and lifestyle were inversely associated with the risk of all-cause mortality among Chinese older adults. We also observed that the overall lifestyle significantly modified the relations between cognitive function and mortality risk, with a healthy diet being the main contributor. The inverse associations between cognitive function and mortality risk were stronger in individuals who adhered to a healthy lifestyle, especially in those with a healthy diet. 

The findings from our data suggest that cognitive function was positively associated with longevity in older populations, which aligns with the findings from several prospective cohort studies in developed countries [7,9,38]. A previous systematic review showed that severity levels of cognitive impairment gave rise to an elevated mortality risk [39]. Similar findings of the inverse relationships between cognitive function and mortality were also found among the oldest Chinese people: 80 years of age and above [40]. In addition, a previous study also suggests that the faster decline in cognitive function was found to be associated with higher mortality independent of initial cognitive function among Chinese older people [36]. 

In line with previous studies [19,20,41], we observed that a combination of favorable lifestyle factors is associated with a lower risk of mortality among older Chinese populations. The selection of healthy lifestyle indicators in this study has been largely guided by prior studies, as these indicators are modifiable and universal. Interestingly, we further found a significant modification effect of lifestyle scores combining the five modifiable lifestyle factors on the associations of cognitive function and mortality. Furthermore, these modification effects remained unchanged after excluding participants who had major chronic disease at baseline. The mechanisms underlying the modification effect of lifestyle on the associations between cognitive function and the risk of mortality remain unclear, while our findings may be partly explained by the close relationship between lifestyle behaviors and cognitive function. Cognitive function might be associated with healthy literacy; as a consequence, people with lower cognitive function are less able to engage in a healthy lifestyle [6,42]. As the potential mechanisms of cognition–death relationships may be partly explained by health literacy, our study adds to the limited evidence examining the association between cognitive function and healthy lifestyles on all-cause mortality risk. Consistently, mounting evidence highlights the importance of healthy lifestyles in reducing the risk of cognitive decline [11,12,35,43].

Among the individual lifestyle behaviors, we found that a healthy diet showed stronger interaction with cognitive function on mortality risk than other lifestyle behaviors. The finding was supported by previous studies that good dietary diversity was associated with a reduced risk of cognitive impairment among elderly people [31,44,45,46]. Even though the biological mechanism underlying the interaction was not clear, several plausible explanations have been proposed. First, good dietary diversity has been reported as a proxy indicator of nutrient adequacy [46] that can help reduce the burden of cognitive impairment [47,48]. Second, low dietary diversity is associated with enhanced oxidative stress, which would affect normal brain function and increase the risk of mild cognitive impairment [49,50]. Third, healthy food diversity is correlated with a more diverse gut microbiota [51], which may influence host cognition via the brain–gut–microbiome axis [52,53]. Further dietary intervention studies are needed to advance the current understanding of the mechanistic effects of dietary modification on cognitive function and mortality.

To be noted, although interactions for other lifestyle behaviors did not reach the statistically significant level, the negative association of high cognitive function with mortality seemed to be more pronounced among individuals with normal weight, no smoking, no alcohol consumption, and regular exercise. Previous studies also showed that smoking, alcohol consumption, regular exercise, and BMI played vital roles in cognitive function [12,54,55]. 

The study has significant implications for the advancement of new public health intervention strategies aiming at improving healthy aging. To our knowledge, the study is the first to assess the interaction between cognitive function and lifestyle on all-cause mortality among Chinese older adults. More importantly, the study also adds to the limited evidence examining the modification effects of a healthy diet on cognitive function and mortality risk. The findings indicate that maintaining a healthy lifestyle could mitigate the adverse association between cognitive function and the risk of mortality, emphasizing the importance of a healthy diet for longevity among the older. These findings, if confirmed by intervention trials, can be taken as a new supplement to personalized health interventions. 

However, the study has several limitations to be addressed. The study was conducted in the CLHLS, in which most participants were Chinese older people. Considering that the model of life varies in China and the rest of the world, the generalizability of our findings to other populations should be interpreted with caution. Future studies in other countries and regions are needed to verify the results. Additionally, although the vast majority of measurable socio-demographic and potential health-related factors were adjusted, some important confounders, including psychological issues and unknown factors, might also cause residual confounding. Moreover, the observational nature of this study precludes the determination of causality, and further randomized clinical trials are necessary to validate our findings about the connections of lifestyles and cognitive function. 

## 5. Conclusions

This longitudinal prospective study indicates that lower cognitive function is associated with an elevated risk of all-cause mortality, and these associations are attenuated by adopting to a healthy lifestyle. Our findings emphasize the importance of considering a healthy lifestyle when investigating the association between cognitive function and longevity. The findings of our study could have significant implications for the development of strategies aimed at promoting healthy aging by enhancing cognitive function in individuals with an unhealthy lifestyle, particularly those with an unhealthy diet.

## Figures and Tables

**Figure 1 nutrients-16-01297-f001:**
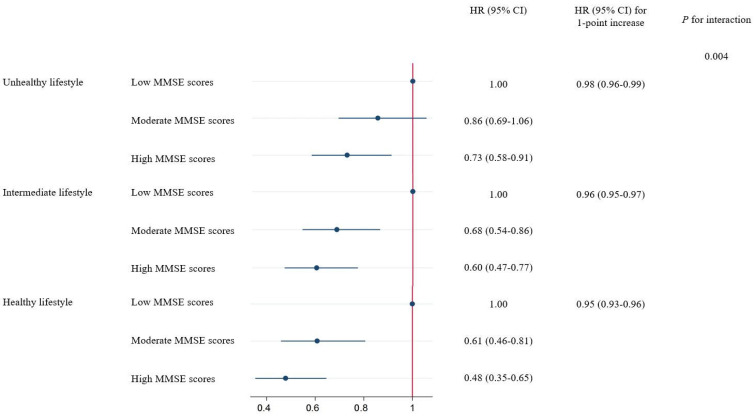
Associations of the MMSE scores with mortality stratified by lifestyles. Associations between MMSE scores and all-cause mortality stratified by lifestyles. Results were adjusted for age, sex, education, residence, marital status, living pattern, self-rated of economic status, ADL in disability, and history of chronic disease (diabetes, heart diseases, cancer, and stroke). MMSE, Mini-Mental State Examination; HR, hazard ratio; CI, confidence interval.

**Figure 2 nutrients-16-01297-f002:**
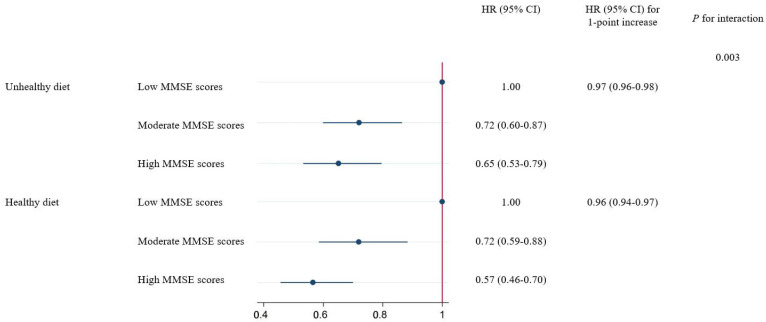
Associations of the MMSE scores with mortality stratified by diet. Associations between MMSE scores and all-cause mortality stratified by diet. Results were adjusted for age, sex, education, residence, marital status, living pattern, self-rated of economic status, smoking status, alcohol consumption, physical activity, BMI, ADL in disability, and history of chronic disease (diabetes, heart diseases, cancer, and stroke). MMSE, Mini-Mental State Examination; HR, hazard ratio; CI, confidence interval.

**Table 1 nutrients-16-01297-t001:** Baseline characteristics of participants according to MMSE scores.

Characteristics	MMSE Score			*p*
	Low	Moderate	High	
*N*	1298	1588	2238	
Age group (%)				<0.001
65–79 years	666 (51.3)	1243 (78.27)	2006 (89.6)	
≥80 years	632 (48.7)	345 (21.7)	232 (10.4)	
Male (%)	365 (28.1)	797 (50.2)	1246 (55.7)	<0.001
No schooling (%)	1015 (78.4)	841 (53.2)	869 (38.9)	<0.001
Urban residents (%)	500 (38.5)	774 (48.7)	1130 (50.5)	<0.001
Married (%)	322 (24.9)	763 (48.2)	1333 (59.7)	<0.001
Living with family members (%)	1027 (79.7)	1247 (78.9)	1798 (81.1)	0.247
Self-rated of economic status (%)				<0.001
Rich	187 (14.5)	292 (18.5)	477 (21.3)	
So-so	842 (65.5)	1058 (67.0)	1524 (68.2)	
Poor	257 (20.0)	230 (14.6)	234 (10.5)	
ADL disabled (%)	346 (27.3)	166 (10.7)	125 (5.7)	<0.001
Heart disease (%)	132 (10.2)	216 (13.6)	286 (12.8)	<0.001
Diabetes (%)	38 (2.9)	74 (4.7)	124 (5.5)	0.001
Cancer (%)	6 (0.5)	6(0.4)	19 (0.9)	<0.001
Stroke (%)	100 (7.7)	117 (7.4)	152 (6.8)	<0.001
Lifestyle (%)				<0.001
Unhealthy	498 (38.5)	609 (38.5)	823 (36.8)	
Intermediate	473 (36.6)	544 (34.4)	698 (31.2)	
Healthy	323 (25.0)	430 (27.2)	715 (32.0)	
Vital status (%)				<0.001
Survived	690 (53.2)	1166 (73.4)	1807 (80.7)	
Deceased	608 (46.8)	422 (26.6)	431 (19.3)	

Categorical variables were presented as *n* (%). MMSE, Mini-Mental State Examination; ADL, activities of daily living.

**Table 2 nutrients-16-01297-t002:** Adjusted HRs and 95% CIs for the MMSE scores with all-cause mortality.

	MMSE Scores			HR (95% CI) for 1-Point Increase	*p* for Trend
	Low	Moderate	High		
Model 1 ^a^	1.00	0.61 (0.54–0.70)	0.49 (0.43–0.56)	0.95 (0.95–0.96)	<0.001
Model 2 ^b^	1.00	0.62 (0.55–0.71)	0.51 (0.44–0.58)	0.96 (0.95–0.96)	<0.001
Model 3 ^c^	1.00	0.72 (0.63–0.83)	0.60 (0.52–0.69)	0.96 (0.96–0.97)	<0.001
Model 4 ^d^	1.00	0.73 (0.64–0.83)	0.61 (0.53–0.71)	0.96 (0.96–0.97)	<0.001

^a^ Adjusted for age and sex. ^b^ Adjusted for age, sex, and lifestyle. ^c^ Adjusted for all the covariates, including age, sex, education, residence, marital status, living pattern, self-rated of economic status, ADL in disability, and history of chronic disease (diabetes, heart diseases, cancer, and stroke). ^d^ Adjusted for lifestyle and all the covariates, including age, sex, education, residence, marital status, living pattern, self-rated of economic status, ADL in disability, and history of chronic disease (diabetes, heart diseases, cancer, and stroke). MMSE, Mini-Mental State Examination; HR, hazard ratio; CI, confidence interval; ADL, activities of daily living.

**Table 3 nutrients-16-01297-t003:** Adjusted HRs and 95% CIs for the lifestyle with all-cause mortality.

	Lifestyle			HR (95% CI) for 1-Point Increase	*p* for Trend
	Unhealthy	Intermediate	Healthy		
Model 1 ^a^	1.00	0.91 (0.81–1.03)	0.69 (0.60–0.79)	0.87 (0.83–0.91)	<0.001
Model 2 ^b^	1.00	0.93 (0.83–1.05)	0.74 (0.65–0.85)	0.90 (0.86–0.94)	<0.001
Model 3 ^c^	1.00	0.95 (0.84–1.07)	0.78 (0.68–0.90)	0.90 (0.86–0.95)	<0.001
Model 4 ^d^	1.00	0.96 (0.85–1.08)	0.80 (0.70–0.93)	0.92 (0.87–0.97)	0.001

^a^ Adjusted for age and sex. ^b^ Adjusted for age, sex, and MMSE scores. ^c^ Adjusted for all the covariates, including age, sex, education, residence, marital status, living pattern, self-rated of economic status, ADL in disability, and history of chronic disease (diabetes, heart diseases, cancer, and stroke). ^d^ Adjusted for MMSE scores and all the covariates, including age, sex, education, residence, marital status, living pattern, self-rated of economic status, ADL in disability, and history of chronic disease (diabetes, heart diseases, cancer, and stroke). MMSE, Mini-Mental State Examination; HR, hazard ratio; CI, confidence interval; ADL, activities of daily living.

## Data Availability

The CLHLS questionnaires are available at http://opendata.pku.edu.cn/ (accessed on 23 April 2024). The full datasets used in this analysis are available from the corresponding author upon reasonable request.

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
