# Peer review of "Cognitive Function, Healthy Lifestyle, and All-Cause Mortality among Chinese Older Adults: A Longitudinal Prospective Study"

_nutrients, 2024, doi:10.3390/nu16091297_

Round 1
Reviewer 1 Report
Comments and Suggestions for Authors
This is a well-designed and carefully organized longitudinal study of a large group of older Chinese men and women that examined the relationship between cognitive functioning and healthy lifestyle on mortality after approximately 6 years. The authors carefully and systematically presented a large amount of data in a manner that would be understandable to the average reader. They also grouped their data to create 4 Models which they independently tested. They found strong inverse relationships between cognitive functioning as measured by the Mini-Mental Status Eaxam and 6-year mortality that was "modified" by a measure of healthy lifestyle in which a healthy diet appeared to be the main driver. "The associations between cognitive decline and increased risks for mortality were attenuated in people with a healthy lifestyle, especially in those with a healthy diet" (line250-251).
I just have just a few other minor comments. I was a little bothered by the fact that the word "diet" appeared in the title when diet was only one component of the lifestyle measure that the authors created. Yet, the measure of a healthy diet appeared to be the major reason that the lifestyle measure significantly interacted with cognitive functioning to impact mortality. Is that correct? I think it would be more accurate and professional sounding if the authors used the term" healthy lifestyle" instead of diet in their title.
I think the Abstract should include the average age of the subjects at the onset of the study.
Minor English problems: Line 9, first sentence in Background: the word "of" should be deleted. Line 34, I believe a word has been omitted as in the sentence that begins with, Cognitive. Cognitive WHAT as fundamental function... ? The meaning of this sentence is difficult to understand. I think they mean that greater than average cognitive decline in older life is a precursor of dementia. Line 69: across not "cross". Line 78: I could not find Figure S1; were supplementary materials included in the application? Line 123: It took me a while to understand that "vital status" was what in English is called "vital statistics". Line 249, the word diet is misspelled. Line 269, a word seems to be left out, "associations between cognitive WHAT and the risk of mortality...". Line 315, should start "This longitudinal prospective study...".
I have one more suggestion for the authors that probably will not make any sense to them. Upon first reading the study, I found myself having to struggle to understand exactly how the interaction between cognitive functioning and lifestyle worked. Sometimes the authors speak of the relationship between cognitive functioning and cognitive decline which can be confusing. Somewhere in the Discussion or Conclusion the authors might want to explicitly define what they mean by "modify" and "augment". Such as: "Higher cognitive functioning when combined with a more healthy lifestyle resulted in a lower risk of mortality". Can they say something that clear and definitive, based on the data they have presented? If so, a statement similar to that possibly in the Abstract and later repeated in the Discussion or Conclusion would assist the general reader who is quickly trying to make sense out of the article.
Finally, in the future the authors might consider formulating their questions in terms of a mediation/moderation model. I believe that the adoption of such an approach would add a powerfully to their conceptual tools.
Comments on the Quality of English Language
The English is generally fine. Every now and then, there is a slip up.
Reviewer 2 Report
Comments and Suggestions for Authors
Discussions about possible correlations between dietary diversity, the level of cognitive functions and mortality are not original. These relationships are the basis of dietetics and the promotion of a healthy lifestyle. The fact that a higher level of awareness and a higher quality of life is the knowledge that allows people to take care of their health through proper nutrition, visits to the doctor, maintaining proper physical condition in old age and intellectual development (reading books, conversations, health education in the media etc.). Considering the specific population (Chinese), psychological issues certainly also play a role, i.e. avoiding stress, the ability to calm down through, for example, meditation and working on yourself. For the above reasons, the work is not particularly original. The cause and effect connection in a general sense is quite obvious - mortality increases in populations with a lower standard of living, poorly nourished, less conscious and, therefore, neglected. Additional factors that were taken into account during the analysis also influence the results in an obvious and expected way, due to their exclusively negative impact on health, cognitive functions and mortality (overweight, smoking, alcohol abuse, physical inactivity). Another issue is the certain endemicity of the demonstrated correlations. The model of life in China, for historical and traditionalist reasons, differs from the model of life in the rest of the world. Therefore, it would undoubtedly be interesting to conduct this type of research, e.g. on the European population.
The authors themselves emphasize that the analysis conducted does not explain the causes and effects of the demonstrated connections. And this is also the expected effect, because it is impossible to go to the detailed level with such general assumptions.
The only thing that attracts attention is the rather strong statistics in terms of the results obtained. A high probability of exposure to increased mortality in the situation of reduced cognitive functions and low standard of living was demonstrated in the work, regardless of the degree of generality of the research conducted. Methodologically, the work is correct and its advantage is certainly the provision of hard evidence on the relationship between diet, a healthy lifestyle and cognitive functions in relation to life expectancy.
Discussion
Lines 252 - 254: the sentence should be corrected regarding the type of correlation found: inverse association is the correct term for cognitive function and mortality. However, the terms cognitive function and longevity used in the sentence characterize, as the work shows, a positive association.
